# Impact of ultraviolet germicidal irradiation on new silicone half-piece elastometric respirator (VJR-NMU) performance, structural integrity and sterility during the COVID-19 pandemic

Thananda Trakarnvanich[1]⊕*, Uraporn Phumisantiphong[1]⊕, Sujaree Pupipatpab[1]⊕, Chayanee Setthabramote[1]⊕, Bunpot Seakow[2,3,4,5]‡, Supanit Porntheeraphat[2,3,4,5]‡, Jakravoot Maneerit[1]⊕, Anan Manomaipiboon[1]⊕

1 Faculty of Medicine, Vajira Hospital, Navamindradhiraj University, Bangkok, Thailand, 2 Digital Agriculture Technology Research Team (DAT), Bangkok, Thailand, 3 Research and Development Intelligent 2Systems Networks (ITSN), Bangkok, Thailand, 4 National Electronics and Computer Technology Center (NECTEC), Bangkok, Thailand, 5 National Science and Technology Development Agency (NSTDA), Bangkok, Thailand

⊕ These authors contributed equally to this work.
‡ These authors also contributed equally to this work.
* thananda@hotmail.com

## Abstract

Since the innovation of our new half-piece elastometric respirator, this type of filtering face-piece respirator (FFR) has been used widely in Thailand. Decontamination methods including ultraviolet C (UVC) germicidal irradiation and 70% alcohol have been implemented to decontaminate these respirators. We then examined the inactivation potential of different decontamination processes on porcine epidemic diarrhea virus (PEDV) and numerous bacterial strains, most of which were skin-derived. To enable rigorous integrity of the masks after repeated decontamination processes, fit tests by the Bitrex test, tensile strength and elongation at break were also evaluated. Our results showed that UVC irradiation at a dose of 3 J/cm$^2$ can eradicate bacteria after 60 min and viruses after 10 min. No fungi were found on the mask surface before decontamination. The good fit test results, tensile strength and elongation at break were still maintained after multiple cycles of decontamination. No evidence of physical degradation was found by gross visual inspection. Alcohol (70%) is also an easy and effective way to eradicate microorganisms on respirators. As the current pandemic is expected to continue for months to years, the need to supply adequate reserves of personnel protective equipment (PPE) and develop effective PPE reprocessing methods is crucial. Our studies demonstrated that the novel silicone mask can be safely reprocessed and decontaminated for many cycles by UVC irradiation, which will help ameliorate the shortage of important protective devices in the COVID-19 pandemic era.

**Data Availability Statement:** All relevant data are within the manuscript and its Supporting Information files.

**Funding:** This study was supported by the grant from Navamindradhiraj University Research Fund (COA 059/63) and grant from The National Science and Technology Development Agency (grant no. FDA-CO-2563-118691-TH).

**Competing interests:** The authors have declared that no competing interests exist.

## Introduction

The demand for personal protective equipment (PPE) and filtering facepiece respirators (FFRs) has unexpectedly and exponentially increased during the COVID-19 pandemic. The ability to disinfect and reuse these devices may be necessary as a shortage mitigation strategy to augment the supply these essential devices.

A variety of techniques for reprocessing N95 respirators have been tested, including autoclaving, the use of steam generated by heat or microwaves, application of ethylene oxide, exposure to vaporized hydrogen peroxide (VHP), immersion in 70% alcohol and exposure to ultraviolet germicidal irradiation [1–4]. The current N95 respirator is composed of filter layers of polypropylene fibers that may be destroyed by decontamination processes. Some of these decontamination methods have been found to be unsuitable for FFR disinfection. As stated by the CDC, the use of autoclaving, 160°C dry heat, 70% alcohol and microwave ovens can degrade FFR filters [5], resulting in failed fit testing and unacceptable levels exceeding the National Institute for Occupational Safety and Health (NIOSH) standards. The most significant concern with reuse of FFRs is the persistence of infectious material on the external surface of the respirators, leading to disease transmission. However, reprocessing policies are now widely accepted as an alternative decontamination method given that the procedures meet certain standards set by the Emergency Use Authorization (EUA) of NIOSH.

The CDC has recommended ultraviolet germicidal irradiation (UVGI), VHP, moist heat and steam as the most promising methods to safely decontaminate FFRs while still preserving the respirator integrity, filtration efficiency and fit capacity [6]. UVGI is a disinfection method that uses short-wavelength ultraviolet C (UVC) irradiation to destroy microorganisms, primarily by cross-linking thymidine nucleotides in DNA and uracil nucleotides in RNA, which alters the DNA/RNA structure and interferes with replication [7]. There are three classifications of ultraviolet light according to wavelength: UVA (320–400 nm), UVB (280–320 nm), and UVC (200–280 nm). Application of UVC is an effective method to decontaminate N95 respirators and is endorsed by the Food and Drug Administration, with good activity against $H_1N_1$, $MS_2$ and *Bacillus subtilis* [4–6]. No significant changes in respiratory material strength or particle filtration efficiency were found when using UVC for decontamination purposes. Vaporized hydrogen peroxide requires gas chambers and ventilation hoods for aeration to keep hydrogen peroxide vapor exposure below 1 ppm prior to use [2]. The limitation of this decontamination method is the risk of toxic chemical residues persisting after treatment. Although treatments with moist heat and steam are highly attractive methods to inactivate viruses on FFR surfaces and have demonstrated minimal loss-of-fit performance after three cycles of application to FFRs, some moist heat is not capable of eradicating highly thermostable pathogens [8] and can result in damage to the FFR component [9, 10]. Steam disinfection includes the use of steam from boiling water and steam sterilizer autoclaves or microwaves. These techniques are limited by their inconsistent generation of steam and potential for sparking [5]. Taking this information into consideration, we chose to evaluate the effectiveness of decontamination by UVC and 70% alcohol in our study.

The study described herein is a continuation of a previously published article. Recently, our team invented a novel silicone mask with a HEPA filter (CareStar and SafeStar, Draeger, Germany) (VJR-NMU silicone half-piece elastometric respirators) [11]; over 4,000 pieces have been supplied all over Thailand during the past three months. We plan to study the ability of UVC and 70% alcohol to decontaminate these novel respirators and investigate the effect of these reprocessing methods on physical degradation of the respirators and biological inactivation of pathogens on the respirators.

## Materials and methods

This study was approved by the Vajira Institutional Review Board COA number 066/256 and was performed with strict adherence to the standards state in the Declaration of Helsinki (October 2000 version). All subjects were given written inform consent before enrollment.

### Silicone masks

We used a silicone half-piece electrometric respirator as described previously. The silicone elastometric mask was made using Silibione MM Series 71791 U silicone (Bluestar Silicones, Shanghai, China) combined with a HEPA filter. Twenty silicone masks were deployed for the test according to the sample size calculation (Fig 1).

### Subjects

Participants were healthcare workers at Vajira Hospital, Navamindradhiraj University, Bangkok, Thailand. The Vajira Institutional Review Board, Faculty of Medicine, Vajira Hospital, Navamindradhiraj University, approved the protocol, and all participants provided written informed consent. The individuals in this manuscript have given written informed consent to publish these case details. The inclusion criteria were healthy volunteers 18 to 60 years old. The major exclusion criteria were contraindications to fit tests, such as asthma, congestive heart failure, anosmia, and ageusia.

### Fit test procedure

All the participants had to pass the initial fit test. After decontamination with UVC irradiation of the masks for 60 min, fit testing was conducted using a qualitative fit test (Bitrex Solution aerosol). We followed the same procedure as described in a previous report [11]. The detail of the participants and test data including face dimension are detailed in S1 Fig and S1 Table and S1 Appendix. The protocol was conducted in accordance with the protocol from the OSHA respiratory protection standard [12], including the number, type, and duration of the exercise, and seal checks in accordance with the manufacturer's instructions [13]. dx.doi.org/10.17504/protocols.io.btxtnpnn

### Sample size calculation for respirator reuse

The sample size calculation is shown in (S2 and S3 Figs). We referred to the research by Lindsley et al. [1], who studied the effect of ultraviolet germicidal irradiation (UVGI) on N95 respirator filtration performance and structural integrity by using 4 types of N95 filtering face piece respirators (FFRs), namely, models 1860, 9210, 1730 and 46727. Since there are no previous studies to cite for references, we used the program G*Power version 3.1.9.4 program to compute the mean difference before and after UVC irradiation. The two groups were compared using a paired t-test. We compared the difference between two dependent means, where $\propto$ = 0.05 (one tail) and the power of the test was 90%. We applied an effect size of 0.8 (large effect size) [14]. The sample size calculated was 19. We then collected 20 silicone masks for this study.

### Cells and viruses

A cross-sectional study was conducted to assess bacterial, fungal and viral contamination in silicone mask respirators from hospital personnel. Identification of bacteria and fungi from the silicone masks was conducted at the microbiology laboratory unit, Central Laboratory Vajira Hospital, Faculty of Medicine Vajira Hospital, Navamindradhiraj University, Bangkok, Thailand.

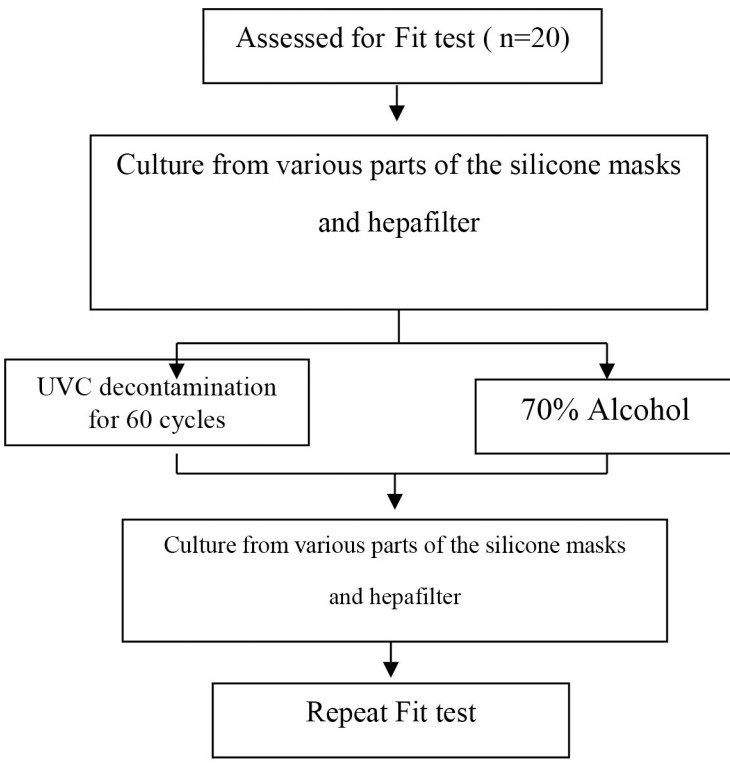

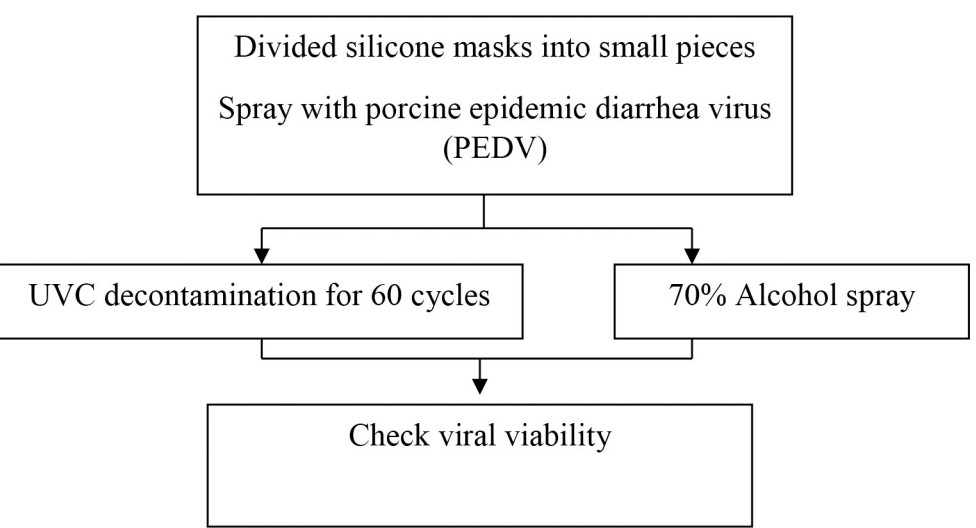

**Fig 1. Study flow chart.**

### Decontaminations of silicone masks

The ability of each decontamination method to inactivate infectious viral particles was measured using inoculated silicone masks. We used porcine epidemic diarrhea virus (PEDV) as

the indicator virus (kindly provided by Dr. Anan Jongkaewwattana from the National Center for Genetic Engineering and Biotechnology, BIOTEC, Thailand). An N99 mask was contaminated by spraying or applying the virus at a concentration of $10^4$ TCID50/ml on the exterior and interior surface of each silicone mask. Additionally, 100 μl viral suspension was spotted onto the filter of N99 masks. Following 20–30 minutes of drying, the silicone masks underwent each of the decontamination procedures, ultraviolet C (UVC) irradiation and 70% alcohol treatment. For UV irradiation, N99 masks were placed in a UVC incubator that emitted a wavelength of 254 nm (UVC, 18.9 mW/cm2) and the masks were exposed for different times: 1, 10 and 20 minutes. For 70% alcohol treatment, the masks were sprayed with 70% alcohol and then dried in a biosafety cabinet. Corresponding untreated control silicone masks were simultaneously spotted with the same viral inoculum and processed for virus detection to account for the effect of drying on virus recovery.

## Effectiveness of decontamination measured by detecting viral infectivity in cell culture

Untreated control masks and treated N99 masks were analyzed for viral infectivity in Vero cell cultures using the microtiter assay procedure. Following decontamination, virus was eluted from the mask material by excision of the spotted areas on each mask and each filter and then transferred into 1 ml virus culture medium (Opti-MEM with 1% penicillin-streptomycin). Each sample was further incubated at 4°C overnight to reconstitute the virus. The elution culture medium was transferred into triplicate wells of Vero E6 cells seeded in 96-well plates. At 48 hours postinfection, cells were examined to evaluate viral infectivity via observation of cytopathic effects. The results for each treatment are expressed as the mean ± standard deviation of 3 biological replicates.

## Sample collection and bacterial isolation

Bacterial and fungal sampling were obtained from the inside and outside surface areas of the masks likely to be frequently touched or in close contact with the face of hospital personnel. Each sample was collected by sterile cotton swabs moistened with 0.85% NaCl, the excess of which was squeezed out of the swabs before swabbing the surface of the samples. Sampling was conducted by the same collector on all occasions to maintain uniformity. Cotton swabs were streaked onto blood agar and MacConkey agar culture media to disperse the microbes to obtain single colonies, and the samples were incubated at 37°C for 24 hours. The HEPA filters inside the respirators were unpacked and immersed in trypticase soy broth (TSB; Difco, USA) (100 ml) in sterile containers and incubated at 37°C for 24 hours. The cultures were then isolated onto blood agar and MacConkey agar. After incubation, the total number of bacterial and fungal colonies was counted, recorded, and selected.

## Bacterial and fungal identification

The bacterial and fungal colonies were identified using Gram staining and biochemical tests. If a colony was found to resemble fungi after the bacteria were isolated, the plate was further incubated at 25°C for 72 hours and 144 hours. The bacteria and fungi were further characterized by MALDI-TOF-MS (autoflex™ speed MALDI-TOF/TOF, Bruker Daltonics, Germany) using flexControl version 3.4. The obtained results were analyzed by comparing the raw spectra with the spectra of the company's library and were expressed as scores. The score ranged from 0 to 3, as recommended by the manufacturer. Score values of >1.7 generally indicated relationships at the genus level, and values of >2.0 generally indicated relationships at the species level. The highest score was used for species identification. The fungal isolates were

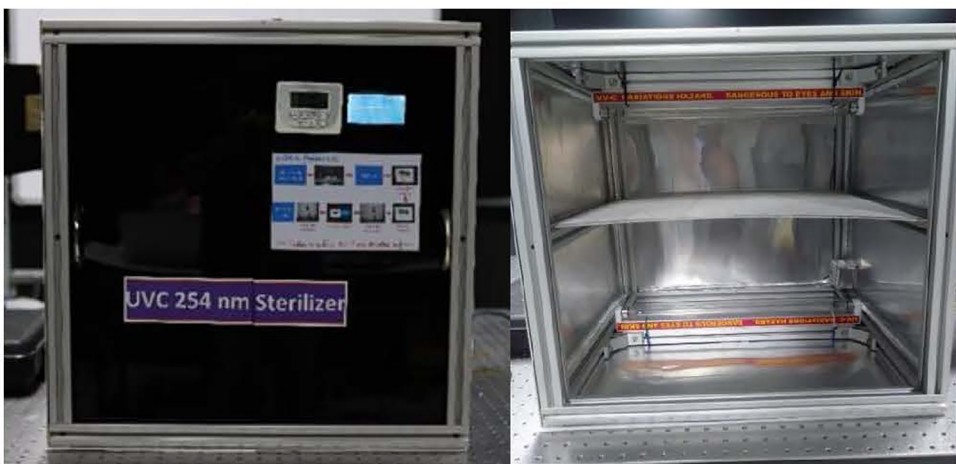

**Fig 2. UVC halogen lamp (Philips UV-C 30 Watt 2 P pieces).**

identified by staining with lactophenol cotton blue (LPCB) containing the following: phenol crystals, 20 g; lactic acid, 20 ml; glycerol, 40 ml; cotton blue stain, 0.05 g; and distilled water, 20 ml [15]. Briefly, a fungal colony was placed on the slide, and a drop of LPCB stain was added. Then, a coverslip was pressed on the material. The LPCB mounts were examined with a light microscope.

After decontamination with UV exposure or alcohol spray for 1, 10, 20 and 60 minutes, the N99 respirators and the HEPA filters inside the respirators were recultivated, isolated and identified by the method described above.

## UVC mask exposure system

We collaborated with the Lighting Platform Spectroscopy and Sensing Devices Research Group (SSDRG) and The National Electronics and Computer Technology Center (NECTEC), Bangkok, Thailand, to help test the physical properties of the new silicone masks after two decontamination methods: dry heat and UVC. The instruments used in this study were as follows:

1. UVC chamber from the Lighting Platform team, SSDRG and NECTEC (Fig 2)

2. UVC meter (International Light IL 1400A Radiometer/Photometer) (Fig 3)

3. Heat bower oven (Fig 4)

4. New silicone half-piece elastometric respirator (Fig 5)

## UVC profile

As depicted in Fig 6, UVC decontamination was conducted using a UV germicidal lamp in a biological safety cabinet. A halogen lamp (Philips UV-C 30 Watt 2 pieces up and down, size: 54.5 cm width, 54.2 cm length, and 54.5 cm height) was used as the UV source. The UVC doses at different parts of the cabinet measured by the UVC meter were more than 300 μWatt-sec/cm$^2$, which is more than the minimal dose to eradicate the RNA virus at 100 μWatt-sec/cm$^2$ for 1 minute (see the supplemental file). In these experiments, UV treatment was applied to all the surfaces at a distance of 22 cm from the light source. We prepared three pieces of silicone straps 10 cm in length (Fig 7) to place inside the cabinet. The UVC decontamination

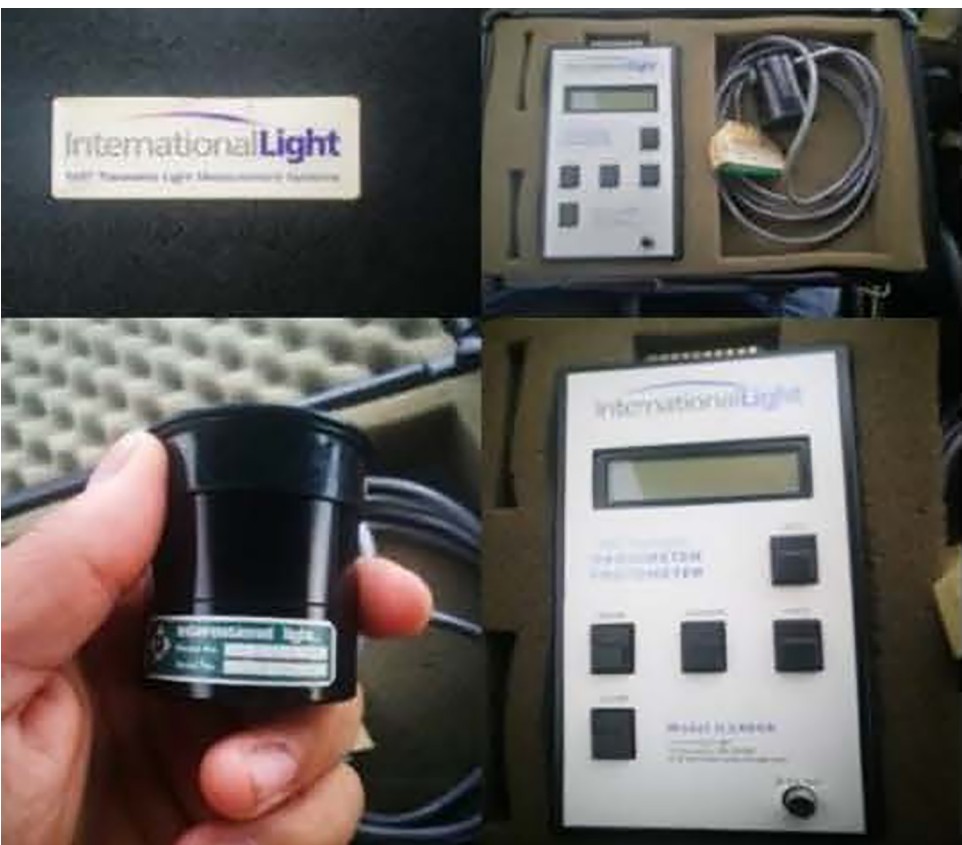

**Fig 3. UVC meter (International Light IL 1400A radiometer/photometer).**

experiment with the silicone mask strap is shown in Fig 8, using dry heat at 75˚C for 5 minutes for 5, 10, 30 and 60 cycles (Fig 9)

## Tensile strength

The tensile strength of the straps was tested by the National Science and Technology Development Agency (NSTDA) and the NSTDA Characterization and Testing Center (NCTC),

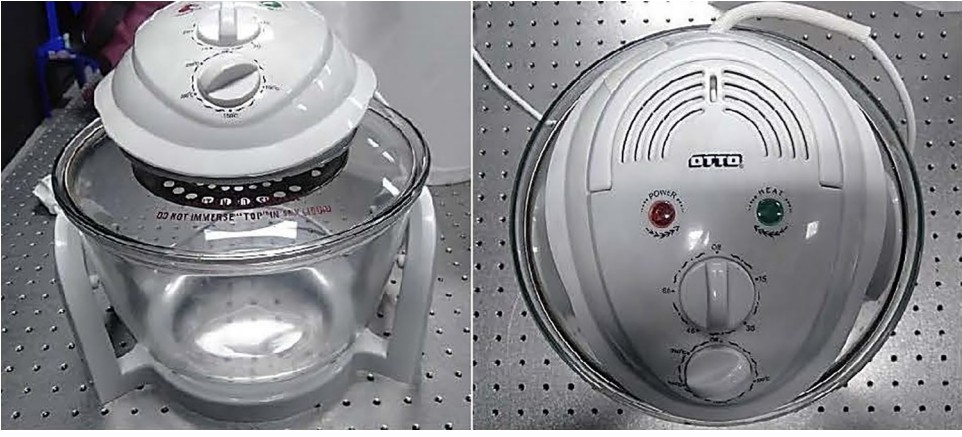

**Fig 4. Heat bower oven.**

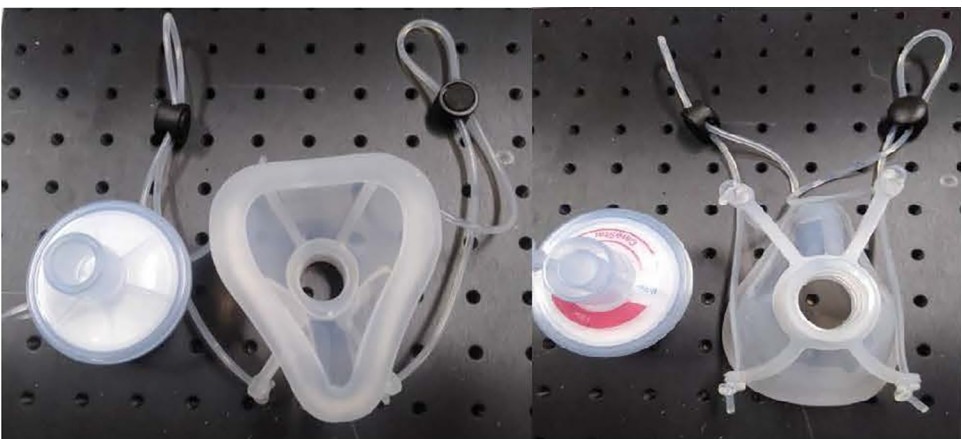

**Fig 5. New silicone half-piece elastometric respirator.**

Bangkok Thailand. Tensile or tension test or tensile test mean texture analysis was performed by using tensile forces that slowly retract and lengthen the material. The increase in tensile force may tear the material apart. We recorded the relationship between tensile stress and tensile strain and then graphically presented the data in a "stress-strain curve", indicating the tensile strain and deformation of the material, which is the distance of stretching from its original shape.

We measured the tensile strength of the silicone strap before and after UVC irradiation to study the strength and resiliency of the strap (Fig 10).

The authors confirm that all related trials for these interventions are registered. The study was registered at ClinicalTrials.gov:NCT04749121. The registration was done after enrolment of participants due to the difficulty of the viral culture in our lab and the possibility of changing the protocol. After adjustment and improvement in nutrient media and uneventful protocol, the registration was performed.

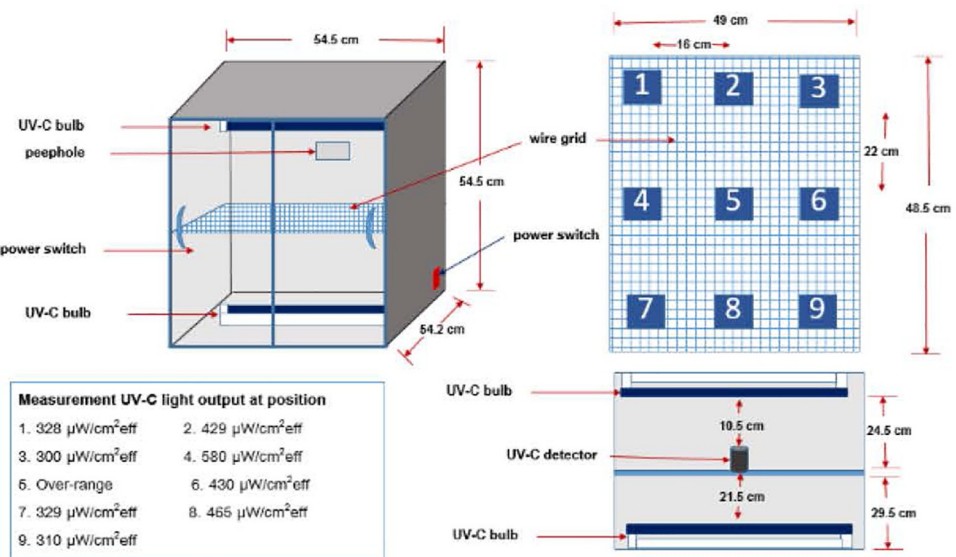

**Fig 6. Lighting platform, measuring UVC light at different locations within the cabinet.**

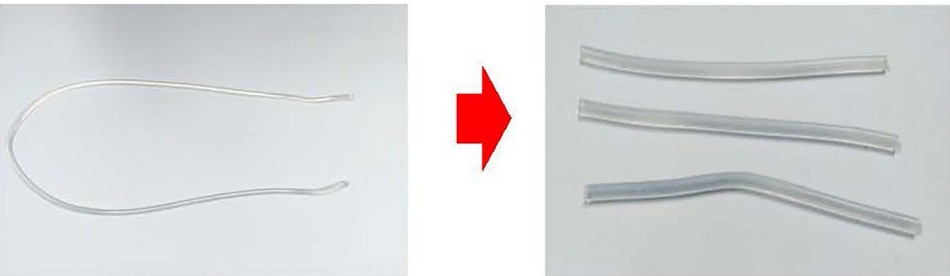

**Fig 7. Preparation of a silicone strap 10 cm in length for decontamination.**

## Results

Our first development of the silicone mask is noninferior to the N99 mask and has been used widely in more than 300 hospitals in Thailand. In December 2020, the rapid and second outbreak of COVID-19 spread all over the country, affecting more than 20,000 people and causing nearly 100 deaths nationwide. Therefore, we tried to find an effective method to protect and contain the pandemic. Safe and effective reuse of our novel respirator might help protect healthcare personnel from this devastating illness.

### Virus extraction

We collected 20 respirators to test the effectiveness of alcohol versus UVC to eradicate viruses and bacteria on the surfaces of the masks, inside the masks and inside the filter. Infectivity was defined as the concentration capable of producing an observable cytopathic effect (CPE) in the cell culture monolayer. After application of 70% alcohol spray on the inner and outer sides of the silicone mask, on the straps and inside the filter, no viral particles were recovered by the CPE method in any of the three areas. After UVC treatment for 1, 10, and 20 minutes, no visible viral recovery was identified after 10 min of decontamination or after 70% alcohol spray. Three replicate tests were carried out for each irradiation time and after alcohol treatment, including the control experiment. There were no viral particles recovered for any of the three cycles tested or in any of the mask locations (inner surface of the mask, outer surface of the mask, strap and inside the filter) (Fig 11).

### Bacteria and fungi

The UVC sensitivity of various microorganisms including bacteria, bacterial spores, and fungi was studied. A total of 59 used masks were collected from 30 healthcare workers to assess

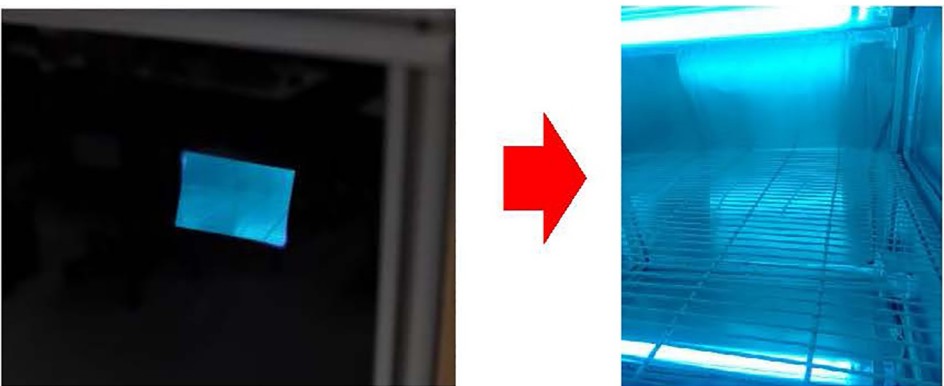

**Fig 8. Decontamination with UVC irradiation at 430 µWatt-sec/cm$^2$ for 1 min for 5, 10, 30 and 60 cycles.**

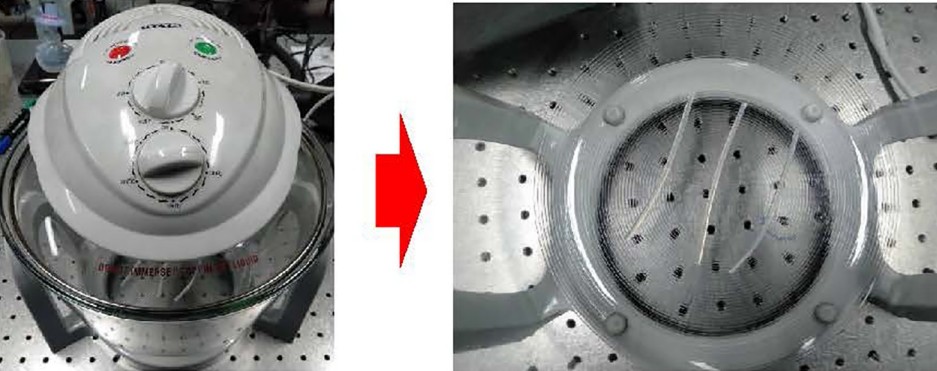

**Fig 9. Dry heat at 75˚C for 5 min for 5, 10, 30 and 60 cycles.**

bacterial and fungal contamination. The used masks were divided into two groups on the basis of the decontamination procedure: UVC treatment (n = 30) and spraying with 70% alcohol (n = 29). The four surfaces of the mask samples (outside, inside, filters and straps) were swabbed and cultured. The bacterial isolates from each part of the mask samples were counted and identified. The results indicate that most bacteria isolated were normal skin flora; however, one of these bacteria can present a risk to human health (Table 1). The most predominant bacteria were *Micrococcus lylae* (20.46%), *Micrococcus luteus* (19.70%), *Bacillus cereus* (17.42%), and *Pseudomonas aeruginosa* (12.88%).

After UVC irradiation for 1 to 20 min, there were residual bacteria inside the filter, on the strap and on the mask surfaces. Almost all of the residual bacteria were normal skin flora (*S. haemolyticus*, *M.luteus*, *B.cereus*). After 60 min of UVC treatment, *P. aeruginosa* and *S. epidermidis* were reduced to 2–3 colonies, as shown in Table 2. We did not find any fungi on the mask, straps or filters in this study (Table 1). A small number of bacterial colonies were found after decontamination with 70% alcohol. The remaining bacteria were *P. aeruginosa*, *S. epidermidis*, *M. lylae* and *M. luteus*, as shown in Table 3.

## Observation of physical degradation

After UVC treatment, there was no obvious change in visible signs of degradation, deterioration, or changes in texture of the material that could be noted and were relevant to the investigation.

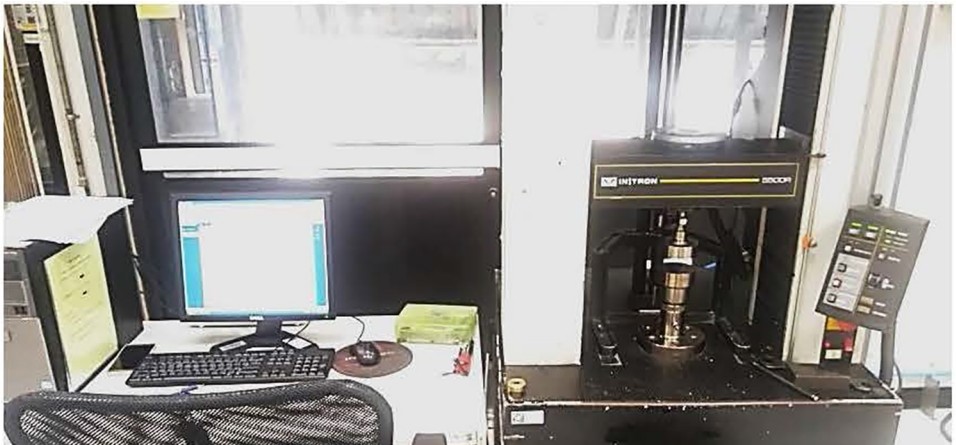

**Fig 10. Tensile system test machine.**

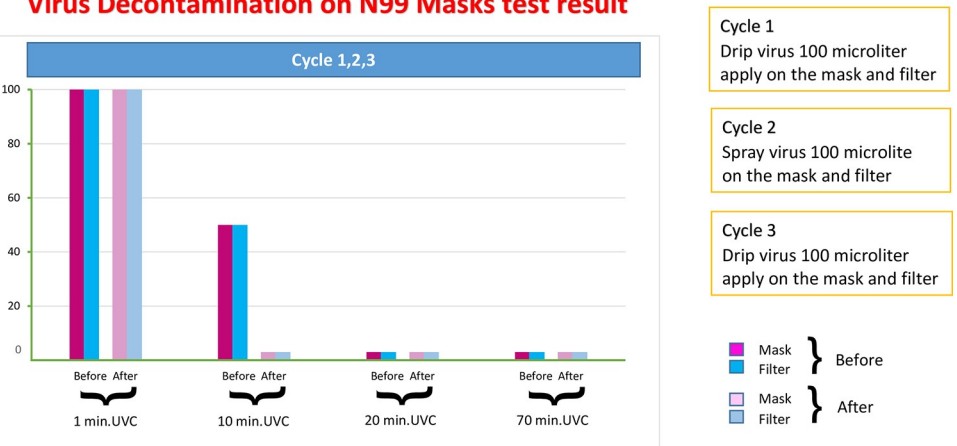

**Fig 11. Effect of alcohol and UVC decontamination on silicone masks at different sites and times.**

## Fit testing

The fit test followed the standard eight exercise step. Table 4 provides a summary of the baseline demographics of all participants. Twenty subjects entered the study (13 male, 7 female), with a mean age of 26.20 ± 4.87 years. Eighteen subjects (90%) passed the first fit test. After adjusting the O-ring strap to tighten the respirators, two subjects passed the second test (100%). The overall fit-test pass rate in this study was 20/20 (100%) (Table 5).

Following the first fit test trial, the masks were irradiated by UVC light for 60 minutes for 60 cycles and returned to the subjects for the second fit test. Table 5 shows the results of the second fit test compared to the control. The fit-test pass rate was 100% even after long-term UVC treatment.

## Tensile strength of the strap before and after UVC irradiation

Table 6 shows that after UVC irradiation, the tensile strength and elongation at break of the strap did not change from the control values, even after 60 cycles of irradiation, which may be

**Table 1. Numbers of microorganisms obtained from 59 used masks from 30 healthcare workers.**

| Microorganism | Number of isolates (CFU/mask) | % |
|---|---|---|
| *Micrococcus lylae* | 54 | 20.46 |
| *Micrococcus luteus* | 52 | 19.70 |
| *Bacillus cereus* | 46 | 17.42 |
| *Pseudomonas aeruginosa* | 34 | 12.88 |
| *Staphylococcus epidermidis* | 24 | 9.09 |
| *Staphylococcus hominis* | 20 | 7.58 |
| *Staphylococcus haemolyticus* | 11 | 4.17 |
| *Kocuria palustris* | 10 | 3.79 |
| *Enterobacter aerogenes* | 7 | 2.65 |
| *Micrococcus luteus* | 2 | 0.75 |
| *Bacillus megaterium* | 2 | 0.75 |
| *Staphylococcus aureus* | 1 | 0.38 |
| *Bacillus cereus* | 1 | 0.38 |

CFU: colony forming unit.

**Table 2. The bacterial isolates and number of colony forming units recovered after UVC irradiation for the specified time.**

| Organism | Before (CFU) | After (CFU) | | | |
|---|---|---|---|---|---|
| | | Outside | Inside | Filter | Strap |
| UVC 1 min | | | | | |
| P. aeruginosa | 24 | 11 | 3 | 2 | 6 |
| M. lylae | 20 | - | - | - | - |
| K. palustris | 5 | - | - | - | - |
| S. epidermidis | 5 | 4 | - | - | 1 |
| S. hominis | 3 | - | - | - | - |
| S. haemolyticus | 2 | - | - | - | - |
| M. luteus | 2 | - | - | - | - |
| B. megaterium | 2 | - | - | - | - |
| E. aerogenes | 2 | - | - | - | - |
| S. aureus | 1 | - | - | - | - |
| B. cereus | 1 | - | - | - | - |
| UVC 10 min | | | | | |
| P. aeruginosa | 20 | 8 | 6 | 2 | 4 |
| M. lylae | - | - | - | - | - |
| K. palustris | - | - | - | - | - |
| S. epidermidis | 4 | 3 | - | - | 1 |
| S. hominis | - | - | - | - | - |
| S. haemolyticus | - | - | - | - | - |
| M. luteus | - | - | - | - | - |
| B. megaterium | - | - | - | - | - |
| E. aerogenes | - | - | - | - | - |
| S. aureus | - | - | - | - | - |
| B. cereus | - | - | - | - | - |
| UVC 20 min | | | | | |
| P. aeruginosa | 10 | 2 | 3 | 2 | 3 |
| M. lylae | - | - | - | - | - |
| K. palustris | - | - | - | - | - |
| S. epidermidis | 3 | 3 | - | - | - |
| S. hominis | - | - | - | - | - |
| S. haemolyticus | - | - | - | - | - |
| M. luteus | - | - | - | - | - |
| B. megaterium | - | - | - | - | - |
| E. aerogenes | - | - | - | - | - |
| S. aureus | - | - | - | - | - |
| B. cereus | - | - | - | - | - |
| UVC 60 min | | | | | |
| P. aeruginosa | 3 | - | 2 | 1 | - |
| M. lylae | - | - | - | - | - |
| K. palustris | - | - | - | - | - |
| S. epidermidis | 2 | 2 | - | - | - |
| S. hominis | - | - | - | - | - |
| S. haemolyticus | - | - | - | - | - |
| M. luteus | - | - | - | - | - |
| B. megaterium | - | - | - | - | - |
| E. aerogenes | - | - | - | - | - |

*(Continued)*

**Table 2.** (Continued)

| Organism | Before (CFU) | After (CFU) | | | |
|---|---|---|---|---|---|
| | | Outside | Inside | Filter | Strap |
| *S. aureus* | - | - | - | - | - |
| *B. cereus* | - | - | - | - | - |

CFU: colony forming unit.

explained by the uniformity of the material from the production process. The elongation at break is the extent to which rubber can be strained before it breaks. After the application of tensile force and stretching of the strap, a percentage of the original length is used to express the elongation at break.

The tensile strength and elongation at break after dry heating at 75°C for 5 min for 1, 5, 30 and 60 cycles did not change from those of the control. The tensile strength after dry heating for 60 minutes differed from the control by only 0.13, while the elongation break increased by 64 units after 60 cycles of the dry heat experiment (Table 7).

## Discussion

During the current COVID-19 pandemic, the supply of N95 masks or filtering facepiece respirators (FFRs) has been limited. Persistent shortages of personal protective equipment (PPE) have driven health care workers (HCWs) to reuse and decontaminate these devices. Recently, our team invented a novel silicone half-piece respirator to help protect healthcare workers and replenish the shortage of N95 respirators during the COVID-19 pandemic. Therefore, it is important and of interest to explore the effective decontamination of this kind of FFR.

We first chose germicidal UVC irradiation as a decontamination method since this method has been studied for a decade [10, 16] and is one of the promising methods identified by the United States Centers for Disease Control and Prevention (CDC) for N95 respirator decontamination [5]. Reductions in bacteria were consistently lower than those in viruses in every area of the masks and filters. A longer UVC decontamination time for as long as 60 min was required to decontaminate most of the bacteria on the mask surfaces. In contrast, 70% alcohol applied on the masks, strap and filters for 10 minutes effectively inactivated the virus and bacteria. No fungi were found at the baseline in the samples. Cadnum et al. [17] found that UVC effectively decontaminated N95 respirators contaminated with Phi6 and MS$_2$ bacteriophages

**Table 3. Bacterial isolates from the masks after spraying with 70% alcohol.**

| Bacterial isolate | Spray with 70% alcohol | | | | |
|---|---|---|---|---|---|
| | Before (CFU) | After 10 min (CFU) | | | |
| | | Outside | Inside | Filter | Strap |
| *M. lylae* | 34 | - | 1 | - | - |
| *M. luteus* | 50 | - | 1 | - | - |
| *P. aeruginosa* | 10 | - | 2 | 1 | - |
| *S. epidermidis* | 19 | - | 2 | - | - |
| *S. hominis* | 17 | - | - | - | - |
| *S. haemolyticus* | 9 | - | - | - | - |
| *K. palustris* | 5 | - | - | - | - |

CFU: colony forming unit.

Table 4. General characteristics of participants (n = 20).

| Parameter | Value |
|---|---|
| Gender, n (%) | |
| Male | 13 (65.0) |
| Female | 7 (35.0) |
| Age (years), mean ± SD | 26.20 ± 4.87 |
| min−max | 21−36 |
| Mask size, n (%) | |
| M | 9 (45.0) |
| L | 11 (55.0) |
| Face length (mm), mean ± SD | 112.55 ± 14.60 |
| min−max | 90−150) |
| Face width (mm), mean ± SD | 113.30 ± 10.13 |
| min−max | 101−140 |

Data are presented as a number (%) or mean ± standard deviation.

and with methicillin-resistant *Staphylococcus aureus* (MRSA). However, the efficacy varied with the different types of organisms and locations on the respirator. This result is not surprising since viruses are smaller than bacteria and contain no cell wall and should be less resistant to UVC light.

O'Hearn et al. [18] reported a meta-analysis of the impact of UVC irradiation on mask efficacy and safety. They found that N95 FFR performance is maintained following a single cycle of UVC irradiation. However, at least one study showed a high fit failure rate after reuse of N95 masks [19]. A study by Ozog et al. [20] showed marked differences in the effects of UVGI on fit testing among various N95 FFR models. While one model was unaffected by a total applied UVC dose of 60 J/cm$^2$, another model failed fit testing after a single 3 J/cm$^2$ dose [21]. Therefore, the findings from one model cannot be extrapolated to others. Lindsley et al. [1] reported the cumulative effect of an extremely high dose of UVC (120 J/cm$^2$) on N95 mask bursting strength to be 11–42%, but this treatment had minor effects on filtration efficiency. We therefore studied the physical degradation and tensile strength of the strap, including fit tests after UVC and 70% alcohol treatment. The tensile strength and elongation at break of our straps were well maintained after these decontamination methods. Dry heat and UVC irradiation did not cause degradation of our silicone masks even after 60 cycles of treatment.

We used the dose range of UVC irradiation that was previously reported to be efficient for bacterial and virus inactivation while maintaining all other key functional respirator properties after multiple reprocessing steps [22]. The tensile strength tests confirmed the durability and strength of the silicone mask, which was easily decontaminated by 70% alcohol. Good levels of FFR fit were retained after multiple decontamination cycles using UVC irradiation. Our study confirmed the safe reuse of novel silicone masks, which could be safely implemented in workplaces.

The principal limitation of this study is that we used only a few decontamination methods. We additionally used porcine epidemic diarrhea virus (PEDV) to simulate SARS-CoV-2; PEDV might not be contagious to humans and may not truly represent SARS-CoV-2, and the viruses might vary considerably in their susceptibility to UVC light. We used UVC doses of 300 μWatt-sec/cm$^2$ or 3 J/cm$^2$, and the exposure falls off by $1/r^2$ from the point source, where r is the distance from the source. The findings in this study are applicable only to the silicone mask and decontamination treatments investigated. Other FFRs may be more easily degraded

**Table 5. Result of fit test.**

| Fit test | Pre UVC | Post UVC | p-value |
|---|---|---|---|
| Number of tests (n), mean ± SD | 17.00 ± 8.01 | 8.45 ± 4.58 | < 0.001[a] |
| min—max | 3–30 | 2–20 | |
| First fit test result, n (%) | | | |
| Fail | 2 (10.0) | 1 (5.0) | 0.564[b] |
| Pass | 18 (90.0) | 19 (95.0) | |
| Second fit test result, n (%) | | | |
| Fail | 0 (0.0) | 0 (0.0) | NA |
| Pass | 20 (100.0) | 20 (100.0) | |

Abbreviations: NA, data not applicable.

[a]Paired samples t-test.

[b]McNemar's test.

**Table 6. Tensile strength of the strap after UV-C irradiation at a dose of 430 μWatt-sec/cm$^2$ for 1 minute.**

| Cycle | Maximum load (N) | Tensile strength (MPa) | Elongation at break (%) |
|---|---|---|---|
| Control | 59.9 | 8.47 | 712.26 |
| 5 | 64.49 | 9.12 | 761.32 |
| 30 | 61.55 | 8.71 | 758.59 |
| 60 | 63.47 | 8.98 | 720.13 |

**Table 7. Mean tensile strength of the strap before and after dry heating at 75°C for 5 minutes.**

| Cycle | Maximum load (N) | Tensile strength (MPa) | Elongation at break (%) |
|---|---|---|---|
| Control | 62.75 | 8.88 | 736.05 |
| 1 | 64.36 | 9.11 | 797.3 |
| 5 | 65.95 | 9.33 | 841.6 |
| 30 | 63.73 | 9.02 | 818.4 |
| 60 | 63.63 | 9.01 | 800.6 |

in terms of physical damage, fitting characteristics and comfort characteristics. We did not test the quantitative dose ranges of UVC by ourselves but used previous data for this determination. Future studies should also assess the relative dose of UVC and other promising decontaminations for this kind of respirator.

## Conclusion

This study showed that UVC irradiation at a dosage of 300 μWatt-sec/cm$^2$ for 1 minute satisfactorily decontaminated the silicone mask, as measured by viral culture. The time to eradicate bacteria was 60 minutes longer due to the tolerance of the bacteria to UVC irradiation. Moreover, 70% alcohol effectively decontaminated the virus and bacteria deposited on the masks and filters. A good level of fit of the masks can be maintained following UVC treatment. The tensile strength of the strap was retained over multiple decontamination cycles using UVC and dry heat for up to 60 cycles of treatment. Given the efficacy of disinfection by UVC and 70% alcohol, these types of decontamination methods could potentially be used for the new type of silicone mask half-piece respirator in the setting of a crisis with inadequate supplies of PPE.

## Supporting information

**S1 Checklist.**
(PDF)

**S1 Fig.** a Bivariate model of face dimension distribution. b Scatter bivariate distribution of face dimension.
(TIF)

**S2 Fig. Sample size calculation.**
(TIF)

**S3 Fig. Sample size.**
(TIF)

**S1 Table. Percentage of population and number of subjects for the panel based on face length and width.**
(TIF)

**S1 File. The experiment by BIOTEC team: Cultured virus on N95 mask by dropping porcine epidemic diarrhea virus (PEDV).**
(DOCX)

**S1 Appendix. Data of participants and investigation results.**
(DOCX)

## Acknowledgments

We are grateful for Dr.Anan Jongkaewattana from the National Center for Genetic Engineering and Biotechnology(BIOTEC) and The National Science and Technology Development (NSTDA) for providing the virus cell line. The team from The National Electronics and Computer Technology Center (NECTEC) and The Research and Development Intelligent Systems and Networks (ITSN). We thank the subjects for participate in this study and Mr. Anucha Kamsom for his statistical analysis.

## Author Contributions

**Conceptualization:** Thananda Trakarnvanich, Jakravoot Maneerit, Anan Manomaipiboon.

**Data curation:** Thananda Trakarnvanich, Anan Manomaipiboon.

**Funding acquisition:** Thananda Trakarnvanich, Jakravoot Maneerit.

**Investigation:** Thananda Trakarnvanich, Uraporn Phumisantiphong, Sujaree Pupipatpab, Chayanee Setthabramote, Bunpot Seakow, Supanit Porntheeraphat.

**Methodology:** Thananda Trakarnvanich, Uraporn Phumisantiphong, Sujaree Pupipatpab, Chayanee Setthabramote, Bunpot Seakow, Supanit Porntheeraphat.

**Project administration:** Thananda Trakarnvanich, Anan Manomaipiboon.

**Resources:** Thananda Trakarnvanich, Bunpot Seakow, Jakravoot Maneerit, Anan Manomaipiboon.

**Software:** Jakravoot Maneerit.

**Supervision:** Thananda Trakarnvanich, Uraporn Phumisantiphong, Sujaree Pupipatpab, Chayanee Setthabramote, Jakravoot Maneerit, Anan Manomaipiboon.

**Validation:** Thananda Trakarnvanich, Sujaree Pupipatpab, Chayanee Setthabramote, Bunpot Seakow, Supanit Porntheeraphat, Anan Manomaipiboon.

**Visualization:** Thananda Trakarnvanich, Uraporn Phumisantiphong, Sujaree Pupipatpab, Chayanee Setthabramote, Bunpot Seakow, Supanit Porntheeraphat, Anan Manomaipiboon.

**Writing – original draft:** Thananda Trakarnvanich, Uraporn Phumisantiphong, Chayanee Setthabramote, Bunpot Seakow.

**Writing – review & editing:** Thananda Trakarnvanich.

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
