## [Decision Letter · Decision Letter 0]

5 Aug 2021

PONE-D-21-10415

Impact of ultraviolet germicidal irradiation on new silicone half-piece elastometric repirator (VJR-NMU) performance,structural integrity and sterility during the COVID-19 pandemic

PLOS ONE

Dear Dr. Trakarnvanich,

Thank you for submitting your manuscript to PLOS ONE. After careful consideration, we feel that it has merit but does not fully meet PLOS ONE’s publication criteria as it currently stands. Therefore, we invite you to submit a revised version of the manuscript that addresses the points raised during the review process.

Please go through the manuscript and revise accordingly.

We look forward to receiving your revised manuscript.

Kind regards,

Prasenjit Mitra, MD, CBiol, MRSB, MIScT, FLS, FACSc, FAACC

Academic Editor

PLOS ONE

Journal Requirements:

3. Thank you for submitting your clinical trial to PLOS ONE and for providing the name of the registry and the registration number. The information in the registry entry suggests that your trial was registered after patient recruitment began. PLOS ONE strongly encourages authors to register all trials before recruiting the first participant in a study.

1) your reasons for your delay in registering this study (after enrolment of participants started);

2) confirmation that all related trials are registered by stating: “The authors confirm that all ongoing and related trials for this drug/intervention are registered

5 PLOS ONE now requires that authors provide the original uncropped and unadjusted images underlying all blot or gel results reported in a submission’s figures or Supporting Information files. This policy and the journal’s other requirements for blot/gel reporting and figure preparation are described in detail at https://journals.plos.org/plosone/s/figures#loc-blot-and-gel-reporting-requirements and https://journals.plos.org/plosone/s/figures#loc-preparing-figures-from-image-files. When you submit your revised manuscript, please ensure that your figures adhere fully to these guidelines and provide the original underlying images for all blot or gel data reported in your submission. See the following link for instructions on providing the original image data: https://journals.plos.org/plosone/s/figures#loc-original-images-for-blots-and-gels. 

Reviewers' comments:

Reviewer's Responses to Questions

**Comments to the Author**

1. Is the manuscript technically sound, and do the data support the conclusions?

Reviewer #1: Yes

2. Has the statistical analysis been performed appropriately and rigorously? 

Reviewer #1: Yes

3. Have the authors made all data underlying the findings in their manuscript fully available?

Reviewer #1: Yes

4. Is the manuscript presented in an intelligible fashion and written in standard English?

Reviewer #1: Yes

5. Review Comments to the Author

Reviewer #1: Thank you very much for your efforts in the manuscript which may add benefits to health caregivers especially in low economic countries to help in the pandemic era

Please i need your explanation about these points

Is it better to use a new HEPA Filter if it is available?

Alcohol 70 % may affect on the effectiveness of HEPA filter

It is difficult to remove accumulated fine particles in HEPA filter

6. PLOS authors have the option to publish the peer review history of their article (what does this mean?). If published, this will include your full peer review and any attached files.

Reviewer #1: No

---

## [Author Response · Author response to Decision Letter 0]

12 Aug 2021

RESPONSE TO REVIEWER COMMENTS AND SUGGESTIONS

Thank you for the opportunity to respond to the editors’ and reviewers’ comments on our revised manuscript. Our responses are provided after each comment. We hope that these extensive changes, based on their critiques, make this manuscript suitable for publication in PLoS one.

COMMENTS FOR THE AUTHOR:

Authors’ response: Thank you for your comments. We have revised the manuscript style according to the journal’s requirement.

2. Please review your reference list to ensure that it is complete and correct.

Authors’ response: Thank you for your comments.We reviewed all the references and found no retracted references. 

3. Your reasons for your delay in registering this study (after enrolment of participants started)

Authors’ response: Thank you for your comments. The registration was done after enrolment of participants due to the difficulty of the viral culture in our lab and the possibility of changing the protocol.After adjustment and improvement in nutrient media and uneventful protocol,the registration was performed. We already described the reason for the delay registration in the manuscript.

4. We note that you have stated that you will provide repository information for your data at acceptance. Should your manuscript be accepted for publication, we will hold it until you provide the relevant accession numbers or DOIs necessary to access your data.

Authors’ response : Thank you for your comments.The repository data was added in the Appendix section in the supplemental file.We will make change to our Data Availability statement in the cover letter.

5.PLOS ONE now requires that authors provide the original uncropped and unadjusted images underlying all blot or gel results reported in a submission’s figures or Supporting Information files.

Authors’ response : Thank you for your comments.Almost all figures in this manuscript are photos taken from our lab.The exception are Fig 11 in the manuscript and S3 Fig in the supplemental file that will be sent together in original files.All of these changes will be mentioned in the cover letter.

6. Please include captions for your Supporting Information files at the end of your manuscript, and update any in-text citations to match accordingly.

Authors’ response : Thank you for your comments.All the captions were added at the end of the manuscript and cited in the text to match accordingly.

 Review Comments to the Author

- Is it better to use a new HEPA Filter if it is available?

Authors’ response

Thank you for your comments If there is enough and available new HEPA Filter, it is definitely better to use the new one.However,due to the high cost of the HEPA filter and short supply,we try to find the easiest and safe way to decontaminate them. 

- Alcohol 70 % may affect on the effectiveness of HEPA filter

Authors’ response

Thank you for your comments.

HEPA filters are made of thin fibers of glass, and contain some level of activated carbon-based material.Therefore it should not be affected by alcohol 70%/ 

- It is difficult to remove accumulated fine particles in HEPA filter

Authors’ response

Thank you for your comments.

If the particulate is dense ,removal will be difficult especially with the HEPA filter from SafeStar (Draeger,Germany).However,our another experiment with new version of our HEPA filter showed that the HEPA filter can be used up to one month and the accumulated fine particles are easily removed

Day Air flow (LPM) Filtration efficiency (%)

1 320.56 99.71

1 320.56 99.81

2 335.83 99.75

3 335.83 99.55

4 335.83 99.38

5 351.09 99.87

6 335.83 99.86

7 335.83 99.94

8 305.30 99.97

9 335.83 99.89

10 274.77 99.67

11 259.50 99.65

12 259.50 99.52

13 259.50 99.39

14 274.77 99.84

15 236.61 99.79

16 224.393844 99.96

17 222.8673553 99.91

18 221.3408665 99.87

19 219.8143778 99.86

20 219.8143778 99.90

21 219.8143778 99.83

22 219.8143778 99.89

23 219.8143778 99.85

24 219.8143778 99.80

25 219.8143778 99.87

26 219.8143778 99.92

27 210.6554454 99.85

28 210.6554454 99.90

29 210.6554454 99.87

30 210.6554454 99.76

 I have removed the previous blot,gel figure since our team used the crop figure and did not save the file.Therefore,this figure did not meet the Journal' s requirement 

I hope this revision will answer and fulfil the reviewer and the Editor’s suggestion.

Sincerely,

Thananda Trakarnvanich M.D.

---

## [Decision Letter · Decision Letter 1]

23 Sep 2021

Impact of ultraviolet germicidal irradiation on new silicone half-piece elastometric repirator (VJR-NMU) performance,structural integrity and sterility during the COVID-19 pandemic

PONE-D-21-10415R1

Dear Dr. Trakarnvanich,

We’re pleased to inform you that your manuscript has been judged scientifically suitable for publication and will be formally accepted for publication once it meets all outstanding technical requirements.

Kind regards,

Prasenjit Mitra, MD, CBiol, MRSB, MIScT, FLS, FACSc, FAACC

Academic Editor

PLOS ONE

Additional Editor Comments (optional):

Reviewers' comments:

Reviewer's Responses to Questions

**Comments to the Author**

1. If the authors have adequately addressed your comments raised in a previous round of review and you feel that this manuscript is now acceptable for publication, you may indicate that here to bypass the “Comments to the Author” section, enter your conflict of interest statement in the “Confidential to Editor” section, and submit your "Accept" recommendation.

Reviewer #1: All comments have been addressed

2. Is the manuscript technically sound, and do the data support the conclusions?

Reviewer #1: Yes

3. Has the statistical analysis been performed appropriately and rigorously? 

Reviewer #1: Yes

4. Have the authors made all data underlying the findings in their manuscript fully available?

Reviewer #1: Yes

5. Is the manuscript presented in an intelligible fashion and written in standard English?

Reviewer #1: Yes

6. Review Comments to the Author

Reviewer #1: No Comments, all comments have been addressed

The authors answered all questions

The reviewer requests have been modified

7. PLOS authors have the option to publish the peer review history of their article (what does this mean?). If published, this will include your full peer review and any attached files.

Reviewer #1: No

---

## [Editor Report · Acceptance letter]

1 Oct 2021

PONE-D-21-10415R1 

Impact of ultraviolet germicidal irradiation on new silicone half-piece elastometric respirator (VJR-NMU) performance, structural integrity and sterility during the COVID-19 pandemic 

Dear Dr. Trakarnvanich:

I'm pleased to inform you that your manuscript has been deemed suitable for publication in PLOS ONE. Congratulations! Your manuscript is now with our production department. 

Kind regards, 

on behalf of

Dr. Prasenjit Mitra 

Academic Editor

PLOS ONE